# Human Maternal-Fetal Interface Cellular Models to Assess Antiviral Drug Toxicity during Pregnancy

Savannah L. Herbek [1,†], Marie C. Smithgall [2,†], Elisabeth A. Murphy [3], Robert E. Schwartz [4], Shuibing Chen [5], Laura E. Riley [6], Heidi Stuhlmann [7], Yawei J. Yang [3,*] and Ria Goswami [1,*]

1   Department of Pediatrics, Division of Infectious Diseases, Weill Cornell Medicine, New York, NY 10065, USA
2   Department of Pathology and Cell Biology, Columbia University Irving Medical Center, New York, NY 10032, USA
3   Department of Pathology and Laboratory Medicine, Weill Cornell Medicine, New York, NY 10065, USA
4   Department of Medicine, Division of Gastroenterology and Hepatology, Weill Cornell Medicine, New York, NY 10065, USA
5   Department of Surgery, Weill Cornell Medicine, New York, NY 10065, USA
6   Department of Obstetrics and Gynecology, Weill Cornell Medicine, New York, NY 10065, USA
7   Department of Cell and Developmental Biology, Weill Cornell Medicine, New York, NY 10065, USA
*   Correspondence: yang@med.cornell.edu (Y.J.Y.); rig4007@med.cornell.edu (R.G.)
†   These authors contributed equally to this work.

**Abstract:** Pregnancy is a period of elevated risk for viral disease severity, resulting in serious health consequences for both the mother and the fetus; yet antiviral drugs lack comprehensive safety and efficacy data for use among pregnant women. In fact, pregnant women are systematically excluded from therapeutic clinical trials to prevent potential fetal harm. Current FDA-recommended reproductive toxicity assessments are studied using small animals which often do not accurately predict the human toxicological profiles of drug candidates. Here, we review the potential of human maternal-fetal interface cellular models in reproductive toxicity assessment of antiviral drugs. We specifically focus on the 2- and 3-dimensional maternal placental models of different gestational stages and those of fetal embryogenesis and organ development. Screening of drug candidates in physiologically relevant human maternal-fetal cellular models will be beneficial to prioritize selection of safe antiviral therapeutics for clinical trials in pregnant women.

**Keywords:** pregnancy; viral diseases; antivirals; maternal-fetal interface; placental ex vivo models; fetal ex vivo models; reproductive toxicity

## 1. Introduction

Viral infections during pregnancy remain a significant health burden, with elevated risks of disease severity for both the pregnant woman and the fetus [1]. While some prenatal viral infections may cause transient maternal symptoms, other viruses can cross the placenta and lead to severe sequelae in the fetus [2,3]. It is estimated that 3% of congenital anomalies are attributable to perinatal infections [4]. Despite these risks, currently, minimal antiviral medications are available as treatment options during pregnancy and pregnant women are mostly excluded from therapeutic clinical trials. To prevent viral disease severity in pregnant women and their fetus, further analysis of the safety and efficacy of existing and new antivirals is needed at the pre-clinical stage. Current guidelines from the U.S. Food and Drug Administration (FDA) recommend embryo-fetal developmental (EFD) toxicity studies in small animal models before any drug proceeds to clinical trials involving pregnant women [5]. While animal studies are important to assess the pharmacokinetics and developmental toxicity of drugs, due to differences in placental biology, metabolism and the genetic landscape of small animals and humans, these models are not always accurate [6–8]. Thus, drug screening in physiologically relevant human ex vivo maternal-fetal interface tissue model systems could be beneficial and the potential use of such models

for toxicity assessment needs to be explored. Here, we review the existing in vitro and ex vivo models available for drug toxicity assessment during pregnancy and the potential benefits and challenges of each model towards their use for antiviral toxicity testing.

## 2. Viral Infections in Pregnancy

Many viruses can lead to increased morbidity and mortality for both the pregnant women and the fetus. Classically a group of infections termed "TORCH" which includes viruses such as cytomegalovirus (CMV), rubella (also known as German Measles), and others such as varicella zoster (VZV), predominantly affect fetal development with varying levels of transmission risk and morbidity depending on which trimester maternal infection occurs [9–12]. Other viruses do not act as teratogens but can still cause adverse fetal outcomes including fetal anemia and hydrops (parvovirus), disseminated disease, central nervous system or skin/mucous membrane disease (herpes simplex) low birth weight, premature labor and spontaneous abortion (rubeola and chikungunya) [13–16].

Poor outcomes during pregnancy can occur outside of TORCH infections. Respiratory infections during pregnancy are also associated with both maternal and fetal risks. During the 2009 H1N1 pandemic, pregnant women had higher rates of complications including bacterial pneumonia and intensive care unit (ICU) admission compared to nonpregnant adults [17]. Neonates born to mothers infected with influenza had increased risk of low birth weight, premature birth and infant death [18]. The effects of other influenza-like-illness causing respiratory viruses such as respiratory syncytial virus (RSV), rhinovirus, coronavirus, parainfluenza viruses 1–4, and human metapneumovirus remain an area of ongoing study [19,20]. However, studies have shown that febrile rhinovirus and human metapneumovirus illness during pregnancy are associated with low neonatal birthweight and increases in pregnant women seeking medical care [21,22].

In the 2010s–2020s, there were outbreaks of several viral infections that had significant impact on pregnant women and the fetus. Zika virus (ZIKV) and CMV causes neurologic complications including neonatal microencephaly [23,24] and studies from the previous severe acute respiratory syndrome (SARS) outbreak indicated that SARS was associated with higher incidences of spontaneous miscarriage, preterm delivery, and intrauterine growth restriction, but without vertical transmission [25]. In 2019 when, a novel coronavirus (SARS-CoV-2) spurred a global pandemic, its effect in pregnancy became a critical question. Data showed that being pregnant is a risk factor for having a severe disease course. Pregnant women had an increased risk of ICU admission, respiratory assistance with ventilation or extracorporeal membrane oxygenation (ECMO) and death compared to similar aged nonpregnant women [26], with increased disease burden among women of color [27,28]. Pregnant women diagnosed with COVID-19 were at higher risk for preeclampsia/eclampsia, preterm birth, stillbirth, and severe neonatal and perinatal morbidity index, compared to uninfected pregnant women [29]. Histopathologic analysis of placentas from women with SARS-CoV-2 infection demonstrated a range of findings including increases in non-specific findings associated with fetal vascular malperfusion, maternal vascular malperfusion or villitis of unknown etiology [30–33]. Rare cases of intra-uterine fetal demise have been reported and characteristic placental findings, termed SARS-CoV-2 placentitis, a triad of chronic histiocytic intervillositis, perivillous fibrin deposition and trophoblast necrosis, have been demonstrated [34–36].This placental damage is likely mediated by complement activation leading to increased cytokine activity and procoagulant effects [35].

## 3. Systemic Exclusion of Pregnant Women from Antiviral Therapeutic Trials

Despite the serious consequences of infection by viruses during pregnancy, for both the mother and fetus, there is a paucity of therapeutics research in pregnancy. This is true for both antivirals as well as medications to treat non-infectious diseases such as hypertension, diabetes, epilepsy, depression and over the counter medications [37–41]. The history of medications leading to devastating consequences warrants this level of stringency due to serious safety concerns for the development of the fetus. In the 1940s

and 1960s diethylstilbestrol (DES) was prescribed to pregnant women to protect against miscarriages and additional pregnancy related complications. However, it was later determined that in utero exposure to DES can lead to clear-cell adenocarcinoma in female offspring and congenital reproductive abnormalities [42–45]. Around the same time in the late 1950s, the sedative thalidomide was utilized to treat nausea during pregnancy, but due to the lack of human safety testing, thousands of infants were born with severe limb malformations [46–48]. In response to such tragedies and in an effort to provide additional research protection for pregnant women, the FDA began to classify them as a "vulnerable" population. Unfortunately, these added regulations perpetuated the exclusion of women from clinical trials by pharmaceutical companies, for decades. The deliberate exclusion of pregnant women from rigorous testing practices fueled the systematic knowledge gap in drug efficacy and safety for these women. Pregnant women, with their physicians, are then faced with challenging decisions weighing therapeutic options, benefits, and potential risks without adequate evidence or data. As a result, both medical providers and patients are often hesitant to use any therapeutics while pregnant, and those that are used are often prescribed "off-label" [49].

It was not until the early 2000's that the Obstetric-Fetal Pharmacology Research Units (OPRU) Network was developed as the framework to study medications used during pregnancy, with a focus on improving the safety and efficacy of therapeutics in pregnant and lactating women. The findings resulting from these studies have demonstrated the direct and clear benefits that maternal-fetal research has for pregnant women and their neonates. In 2017, the Task Force on Research Specific to Pregnant Women and Lactating Women was established and its 2018 report to Congress recommended expanded research to address the severe limitations in scientific evidence on safety, effectiveness, and dosing of therapeutics for pregnant and lactating women [50].

Systematic studies on the safety of therapeutics in pregnancy are difficult to establish not only for existing medications but especially for novel therapeutics. This need is ever more evident in emergency settings such as during the SARS-CoV-2 global pandemic. One review early in the COVID-19 pandemic compiled the evidence of placental transfer and pregnancy safety data for SARS-CoV-2 therapeutic candidates [51]. Very recently, a comprehensive review outlined the safety and efficacy among pregnant patients of an FDA-approved SARS-CoV-2 drug, Remdesivir, from observational studies [41]; though at the height of the pandemic no such data existed. Current evidence on the effect of other antiviral therapeutics in pregnancy is also minimal. After the 2009 H1N1 influenza pandemic, the Roche oseltamivir (Tamiflu) safety database was studied, and the subsequent analysis found that oseltamivir is unlikely to cause adverse pregnancy or fetal outcomes [52]. However, other antivirals such as protease inhibitors used to treat HIV are associated with adverse perinatal outcomes including small for gestational age birthweight [53]. Overall, data on drug safety in pregnancy is derived predominantly from retrospective studies that look at birth and child development outcomes in children with in utero drug exposure. As new antiviral therapeutics are being developed for newly emerging infections or pre-existing viral pathogens, establishment of research tools to prospectively screen safe antivirals for pregnant women should be a priority.

## 4. Human Maternal-Fetal Interface Models to Screen Safe Antiviral Drugs

According to the current U.S. FDA guidelines, a drug candidate must be assessed for EFD toxicity in two animal models, typically one rodent and one non-rodent (rabbit), before being included in human clinical trials involving pregnant women [54]. However, in several previous EFD studies, the drug toxicological profiles estimated using these small animal models did not accurately recapitulate to those observed in humans. The most important example in this context is the drug thalidomide, that resulted in thousands of birth defects and fetal deaths [55], despite being non-teratogenic in EFD studies in mice and rats [56]. In contrast, while penicillin is commonly prescribed in pregnancy [57], this drug is toxic in guineapigs and hamsters [58]. Additionally, pharmacokinetic and phar-

macodynamic changes during pregnancy is significantly different in humans, compared to small animals [59]. While rhesus macaque placental phenotype and genetic makeup is closer to humans, this model also lacks molecular translatability [60], highlighting the need for a preliminary screening of antiviral drug toxicity in human-specific cellular models, along with in vivo EFD toxicity data in animals.

During gestation, humans develop a complex hemochorial placenta [61], an extraembryonic organ formed from both maternal and fetal cells. The placenta supports the developing embryo by exchanging nutrients, gas and waste materials between the mother and the fetus [62], but also allows drugs administered to pregnant women to cross the placenta and be detected in fetal blood flow [63]. Hence, the development of ex vivo models to assess therapeutic toxicity during pregnancy will involve performing drug toxicokinetic studies on models of both the placenta (maternal) and the developing embryo (fetus) (Figure 1). Here, we will discuss the models that represent the placenta, embryo, and fetal organs, including their advantages and limitations and the implications of these models in evaluating the reproductive toxicity of antiviral drugs.

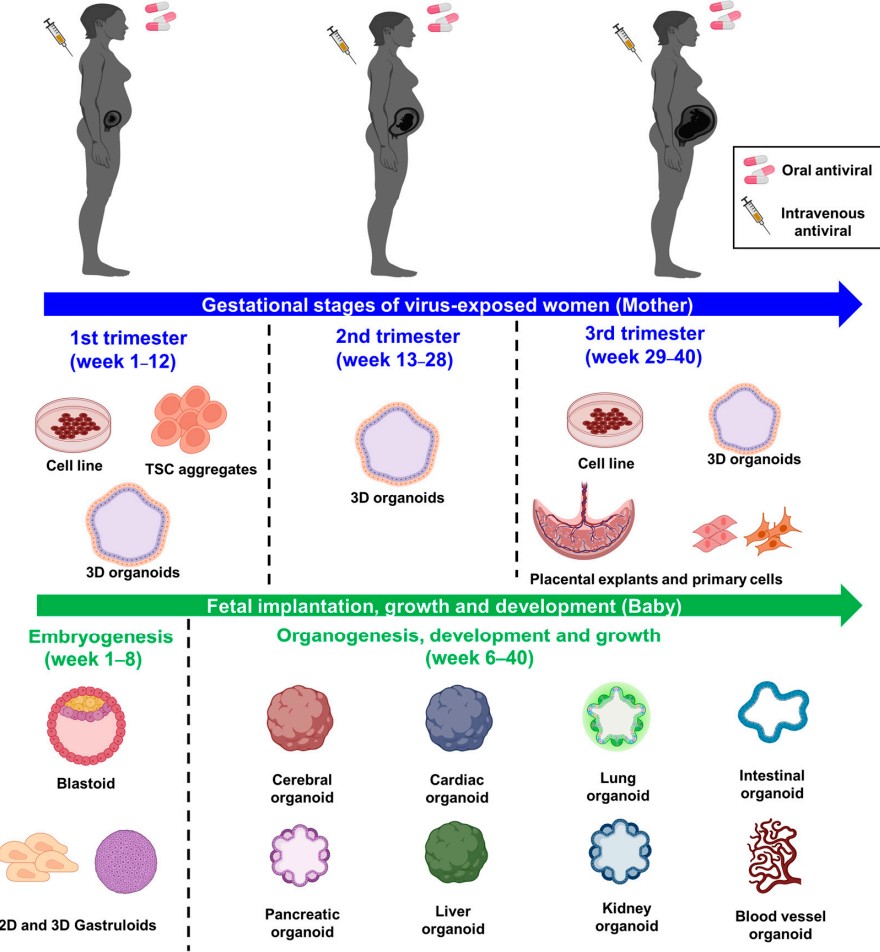

**Figure 1.** Cellular models available to test the antiviral drug toxicity and teratogenicity at the maternal-fetal interface. To develop safe antiviral drugs for treating virally infected pregnant women, preliminary screening of drug toxicity at the maternal-fetal interface is crucial. To assess the toxicokinetic profile of the drug at different gestational stages of the placenta, several 2D and 3D models, such as immortalized cell lines, placental explants, primary cells, trophoblastic stem cell (TSC) aggregates and organoids are available. For antiviral drugs that can cross the placenta and come in contact with the developing fetus, cellular models of embryogenesis (blastoids and gastruloids) and organogenesis and development (organoid models representing fetal organs) can be used to monitor drug toxicity and teratogenicity. Figure is constructed using Biorender.

### 4.1. Cellular Models Representing Human Placenta

The placenta originates from the trophectoderm (TE) cell layer, the outermost epithelial layer of blastocysts. Once the blastocyst adheres to the maternal endometrium (which becomes the decidua), the TE layer fuses to form a primary syncytial layer. Following implantation, the primary syncytial trophoblasts (SCTs) invade through the epithelial cell layer of the maternal endometrium and is eventually completely embedded. As pregnancy progresses, the proliferative cytotrophoblast (CT) cells located beneath the SCT break through the syncytial layer and forms the chorionic villous structures, which are surrounded by intervillous space. As the villous tree continues to develop and enlarge, extraembryonic mesenchymal cells penetrate the villous core and fetal capillaries are developed. Furthermore, villous cytotrophoblast (VCT) cells penetrate through the SCT and merge laterally to form the continuous cytotrophoblastic shell between the chorionic villi and the decidua. CTs in contact with maternal decidua leave the shell and invades the decidua as extravillous trophoblasts (EVT). The placenta continues to develop and mature as pregnancy continues and contact between maternal blood and placenta is only established after conversion of maternal spiral arteries by EVT and development of the hemochorial placenta [62]. While trophoblasts constitute the major cell type of placenta, other cell types are also found at the maternal-fetal interface that includes, fetus-derived macrophages (Hofbauer cells) [64,65], fibroblasts [66], NK cells [67], and T cells [68,69].

With the recent advances in tissue culture and regenerative medicine, several 2-dimensional (2D) and 3D models representing major cell types of the human placenta have been developed that includes placental cell lines, placental primary cells and explants, human stem cell-derived trophoblasts lineages, placental organoids and engineered placental 3D models (Table 1, Figure 1). While several of these models are already being used for therapeutic toxicity assessment, the potential for future applications is vast.

#### 4.1.1. Placental Cell Lines

Placental cell lines, especially trophoblast-like cell lines have been developed for studying placental abnormalities and replication kinetics of pathogens. Three groups of cell lines have been widely used; trophoblast cells derived from choriocarcinomas (BeWo [70], JAR [71], JEG-3 [72]), immortalized first-trimester placental cells (Swan71 [73], ACH-3P [74], HTR8/SVneo [75], 3A- subE [76]) and third trimester hybridoma cell lines AC1-M59 [72]. Even though these cell lines are cancer cell lines or immortalized in nature, they were found to release placental hormones [71], express glucose transporters as observed in human placenta [77], demonstrate placental barrier functions [78] and have trophoblast migration properties [79]. Such trophoblastic cell lines were used as infection models for ZIKV [80], a virus that can cross human placenta [81]. Moreover, these cell lines were used to test the protective efficacy of experimental ZIKV antivirals such as Palmitoleate [80] and non-nucleoside RNA Polymerase inhibitors [82]. In one study, the effect of ZIKV antiviral Palmitoleate on trophoblastic apoptosis and endoplasmic reticulum stress was investigated [80]. Placental trophoblastic cell lines were also used for estimation of the cytotoxicity of endocytosis and efflux inhibitors [83], apoptosis [84], generation of reactive oxygen species (ROS) induced by anti-depressants [85] and anti-tumor drug-mediated inhibition of cellular proliferation [86].

The major advantages of using these cell lines are their wide availability, reproducibility, ease and cost effectiveness of culturing, including their permissiveness to genetic manipulations. However, the genetic signatures of these cell lines are very different from normal trophoblasts [87], and hence the physiological relevance of the measures of cytotoxicity or genotoxicity in these cells will need further investigation. Additionally, there is significant variability between each of these cell lines in functional parameters such as barrier functions and endocrine properties. Moreover, these cell lines have limited-to-no capacity of differentiation and cannot be used to assess the impact of therapeutics in the setting of the morphologically changing placenta through multiple gestational ages. Hence,

whether these cell lines could be routinely utilized as research tools for primary screening of antiviral toxicity in the setting of pregnancy needs further research.

**Table 1.** Advantages and disadvantages of cellular models of human placenta for antiviral toxicity research.

| Cell Model | Advantages | Disadvantages |
|---|---|---|
| Placental cell lines  | • Widely available<br>• Reproducible<br>• Cost effective | • Derived from cancer cell lines or immortalized cells<br>• Genetic signatures different from normal trophoblasts |
| Isolated primary trophoblasts  | • Representative of early trophoblast development<br>• Ability to study cell type-specific drug toxicity. | • Limited accessibility<br>• Labor intensive to isolate<br>• Limited proliferative capability in culture |
| Trophoblastic stem cells (TSCs)  | • Can be differentiated further<br>• Representative of early trophoblast development<br>• Reproducible | • Limited viability post differentiation<br>• Distinct from primary trophoblasts |
| 3D placental organoids  | • 3D structure with cell–cell interaction<br>• Contains mixture of different cell types | • Anatomically inaccurate or "inside-out"<br>• Lacks placental immune system components |
| Placental explants  | • Most physiologically and anatomically comparable to in vivo conditions<br>• Maintain placental function, transport, drug/nutrient metabolism | • Short viability window<br>• Limited accessibility to early term placentas |

### 4.1.2. Isolated Primary Trophoblasts

Primary trophoblast cells such as VCT [88,89] and EVT [90] isolated from placentas can retain functional characteristics of the placental tissues. In fact, the isolated VCT cells could be cultured in vitro and differentiated to SCTs, that can express placental hormones [89]. Isolated placental trophoblast cells were mostly found to be resistant to viral infection, in vitro [91], mimicking human placenta, although infection of cultured primary trophoblast cells with ZIKV was demonstrated [92].

Although, isolation of pure cell populations is labor intensive and the placenta-specific ultrastructure and cellular microenvironment is not maintained, one of the advantages of using such purified cell populations is the ability to study the effects of therapeutics on cell type-specific toxicity. Unfortunately, isolated trophoblasts are difficult to obtain and rapidly lose proliferative capacity in culture media and hence cannot be used to study the effect of therapeutics for long-term on placental trophoblast differentiation process.

### 4.1.3. Trophoblastic Stem-Cell (TSC)-Derived Trophoblast Models

Another approach to study early trophoblast is by using human embryonic stem cell- (hESC) or human inducible pluripotent stem cell- (hiPSC)-mediated trophoblast differentiation [93,94]. Trophectoderm cells and early trophoblastic lineages obtained by

differentiation of hESCs have been used to study infection kinetics of viruses such as ZIKV [93,95] and SARS-CoV-2 [96].

Unlike isolated primary trophoblasts, cultures of TSCs isolated from first trimester placental explants or human blastocyst outgrowths [97] can be differentiated to SCTs or EVTs. Additionally, these cell lines can be used to study the toxicity of antiviral drugs during early pregnancy as TSCs can be reprogrammed into induced trophoblastic stem-like cells (iTSC) which have similar transcriptomic profile compared to first trimester placenta [98]. However, these models are still short-lived when grown in 2D culture and do not fully represent primary trophoblasts or the 3D architecture and cell–cell interactions [99,100].

### 4.1.4. Placental Organoid and Engineered 3D Models

Over the past several years, there have been rapid advances in 3D culture systems with the development of miniature organs or organoids to create biologically relevant ex vivo models of the placenta. For development of placental organoids, three different approaches were taken; (i) development of self-replicating trophoblast organoids [101], (ii) human trophoblastic stem cell (hTSC)-derived organoids isolated from first trimester placental explants [102] and (iii) hiPSC-derived trophoblast organoids [103]. In addition to organoid models, several engineered 3D placental models such as rotating wall vessel bioreactors [104,105], placenta-on-a-chip [106], spheroids of placenta-derived mesenchymal stem cells [107] and hiPSC-derived placental bud models [108] have been constructed, which can be instrumental research tools to study host-pathogen interactions, placental biology and drug toxicology.

Self-replicating trophoblast organoids were established from proliferative VCT cells isolated from early [101,109], mid gestation or term placental explants [110]. These organoids can be cultured long term ex vivo while maintaining their genetic stability. A high degree of correlation was observed between these organoids and explant-isolated placental villi [101] or primary VCT preparations [109] and their complex structure closely resembles the organization of placental villi in vivo. In addition, they are endocrinologically and metabolically similar to in vivo placentas. Finally, the EVT cells in this trophoblast model showed invasive potential [101]. Recently, hTSCs isolated from blastocysts and first trimester CTs generated 3D trophoblast organoids with similar architecture, cellular composition and functions including EVT differentiation as primary trophoblast organoids [102]. With recent advancements in development of these 3D organoid models, their applicability in reproductive toxicity research warrants further investigation. hTSC-derived trophoblast organoids have been used to investigate the infection and replication of ZIKV and SARS-CoV-2 [102].

A major advantage of these organoid-based 3D models is the ability to represent different gestational ages. Moreover, in 3D models, placental tissue morphology, cell-matrix interactions and placental functions could be kept intact even after multiple passages, and hence the toxicological outcomes of experimental therapeutics are more physiologically relevant than 2D models. However, these organoids are cultured "inside-out", with the SCT cells being at the center of the organoids. Hence, to truly mimic viral exposure or antiviral drug treatment, microinjection of the virus or antiviral drug to the internal SCT layer might be necessary.

### 4.1.5. Placental Explants

Primary placental explants cultures offer the most accurate representation of the 3D placental tissue. Placental explant models have been used for assessment of placental functions, transport and nutrient metabolism. Placental explants have been used as models for several infectious diseases, such as human CMV [111], ZIKV [112,113] and SARS-CoV-2 [114]. Furthermore, these models were used to study the toxicology of therapeutics, such as anti-cancer drugs [115], cholesterol medications [116] and pregnancy complications [117]. Additional placental perfusion models allow us to study the transport of drugs or drug metabolites between maternal and fetal circulation [118–121].

Placental explants are more physiologically relevant to assess drug toxicity than placental cell lines, as the microarchitecture and cellular composition of the tissue remains intact. In addition, tissue explants also maintain cell–cell interactions and paracrine communications that results in a more reliable assessment of the effect of a toxic drug on parameters such as metabolism, hormonal production, etc. However, the major limitation of placental explant cultures is the significantly short lifespan for which they can be cultured ex vivo and the inaccessibility around collection of placentas from different and especially earlier gestational ages. To address the issue of short lifespan of placental explants, cryopreservation techniques of villous explants have been utilized [122]. These cryopreserved placental cells were demonstrated to maintain placental morphology and integrity [123] and hence could be used as cellular models for antiviral toxicity testing.

### 4.2. Cellular Models Representing the Embryo and Fetal Organ Development

While placental models could be used to assess maternal toxicity and transplacental drug transfer, models of the developing fetus are necessary to holistically ascertain reproductive toxicity of antiviral drugs such as ganciclovir that are well established to cross the placenta and detected in fetal tissues [124]. Hence, to assess the potential toxicity profiles of newly developed drug candidates, assessment of toxicity in cellular models of the developing embryo and fetal organs is crucial. In humans, after fertilization of the oocyte, the zygote divides rapidly and differentiates into the blastocyst phase of the embryo. Post-implantation of the blastocyst into the mother's uterus, the embryo enters the gastrulation phase with the development of the three germ layers: endoderm, mesoderm, and ectoderm [125]. Subsequently, these germ layers undergo organogenesis by differentiating into a multitude of tissue types [126]. Early embryonic development can be studied with stem cell-based embryonic cultures and blastocyst and gastrulation models, while organ specific cellular models offer deeper insight into later stages of fetal development. Many of these models offer the potential to study drug toxicity with both cell type and temporal specificity (Figure 1).

### 4.2.1. Cellular Models of Early Embryonic Development

To study early human embryo and test drug toxicity in this cellular model, stem cell-based cultures have been developed [127,128]. To better model the 3D structure of human embryo, embryoid bodies that spontaneously differentiate from stem cells and are cultured in suspension have been utilized by the teratology field [129,130]. Recently, these embryoid bodies were used for toxicity evaluation of 33 approved pharmaceuticals and 12 proprietary drug candidates. Additionally, hESC aggregates were cultured with developmental signal modulators, that resulted in morphologically and molecularly similar tissue to paraxial mesoderm and neuroectoderm [131]. This culture system was used to assess the toxicity and transcriptional alterations due to treatment with 18 pharmaceutical drugs [131].

To model the blastocyst phase in human embryonic development, several groups have generated stem-cell derived blastocyst-like cultures or blastoids [132–137]. This has been achieved through aggregating naive pluripotent stem cells [132–134], extended pluripotent stem cells [135,136], or reprogrammed fibroblasts [137]. Single-cell transcriptomic analysis revealed that several of these cellular models closely mimic human blastocysts [138]. When these blastoids were cultured for extended periods of time, they attached to the culture dish, mimicking the behavior of ex vivo embryo cultures [134,137]. The potential of these blastoids to mimic the implantation phase has also been investigated by seeding them on a 2D open-faced endometrial layer, stimulated with hormones, and observing attachment and interaction of the blastoids with the endometrial layer [132]. Recently, hESCs derived from human blastocysts was used to assess the reproductive toxicity of a SARS-CoV-2 antiviral drug remdesivir (RDV) and its metabolite GS-441524. In this study, the survival and proliferation of hESCs treated with RDV or GS-441524 was assessed and the ability of

RDV, but not GS-441524, to decrease the proliferation of cells in a dose-dependent manner was indicated [139].

To model the gastrulation phase of human embryonic development, 2D micropattern and 3D gastruloid cellular models have been described. Micropattern-based teratogenicity assays have been used to assay the dose-dependent cytotoxicity of pharmaceutical compounds, including teratogenic drugs such as thalidomide [140,141]. A major advantage of this model is that the cell aggregates are easily imaged and have a regular shape that makes high throughput toxicity screening feasible. In contrast to 2D models of gastrulation, 3D gastruloids have the advantage of modeling the embryo in three dimensions and have been demonstrated to mimic elements of gastrulation such as formation of the body plan and axial extension [142,143]. A 3D aggregate of hESCs has been shown to differentiate into the three germ layers and faithfully recapitulate their spatial organization [144]. These 3D gastruloid models have been used for reproductive toxicity studies. Notably, 3D gastruloids treated with retinoic acid, a compound known to disrupts gastrulation and cause birth defects, demonstrated disruption in gastruloid elongation [144]. Furthermore, human gastruloid models reliably evaluated the toxicity of other previously established teratogenic compounds [145].

The scalability and cost effectiveness of the stem cell-based embryonic cultures allow for high throughput drug toxicity screening. However, one major disadvantage of these models is their inability to test multiple embryonic lineages and failure to recapitulate the spatiotemporal dynamics and morphology of the gastrulating embryo [146]. In contrast, blastocyst and gastrulation models maintain morphological characteristics of the developing embryo and allow for the assessment of the earliest period of embryogenesis, which may be harder to recapitulate in other culture conditions. However, these models cannot further differentiate and lack the maternal–fetal crosstalk. Therefore, these in vitro and ex vivo models are not entirely translatable and might not serve as standalone models for human reproductive toxicity testing. To overcome these challenges a combination of preliminary toxicity screening in these human models paired with validation in animal models might be optimal.

### 4.2.2. Cellular Models of Organ Development

To study the development of multiple organs and tissues from the three embryonic germ layers, hiPSC- or hESC-derived organoid models, representing fetal tissues and organs, have been developed (Figure 1). These models have been and can be further utilized for studying viral infections and assessment of drug toxicity on both development as well as function. While almost any organ of the developing fetus can be impacted by the toxic effects of drug candidates, stem cell-derived organoid models of specific organs including brain, heart, lung, and kidney have been extensively studied and characterized in the setting of viral infections and drug toxicology. These organoid models were used to screen non-toxic antiviral drugs by evaluating their effects on cell survival, proliferation, and apoptosis [147–149].

Cerebral organoids that can model different regions of the brain and recapitulate aspects of fetal brain development have been widely studied [150–154]. Recently, iPSC-derived cerebral organoid models have been used to study replication patterns of viruses, including CMV, ZIKV and SARS-CoV-2 [149,155–157]. hESC-derived midbrain and hindbrain models and human cortical organoid-on-a-chip models were used for testing the teratogenicity of valproic acid and isotretinoin [153,158]. Furthermore, these organoid models were used for studying antiviral efficacy and toxicity [148,149,159,160]. The effect of antiviral drugs and other therapeutics on cerebral organoid models that can mimic brain development [154] or functionality [161] will be crucial to assess developmental toxicity profiles of drugs.

hPSC-derived cardiac organoids or cardioid models that can form heart chamber-like architecture has been described [162]. Cardiac organoid models have validated the compartment-specific effects of aspirin, thalidomide and retinoid derivatives known to

cause congenital defects [162]. Similar toxicity studies with recalled drugs and other non-toxic compounds were conducted using iPSC-derived cardiac organoids [163].

The efficacy of antiviral drugs for respiratory infections have been studied recently using lung organoid models [164]. Several groups have investigated the permissiveness of lung organoids to SARS-CoV-2 infection [164–170]. Importantly, drug repurposing studies have tested antiviral efficacy of FDA-approved compounds in SARS-CoV-2-infected hPSC-derived lung organoid models [164]. In another study, the cytotoxicity of a combination therapy of IFN$\alpha$2a and remdesivir was assessed in a lung organoid model of SARS-CoV-2 infection [147].

Kidney organoid models have been used to test the efficacy and cytotoxicity of clinical-grade human recombinant soluble angiotensin converting enzyme 2 (hrsACE2) against SARS-CoV-2 infection [171]. Several additional organoid models have been developed such as hPSC or iPSC-derived human blood vessel organoids [172], intestinal organoids [164], liver organoids [173,174] and pancreatic organoids [175], all of which could be instrumental tools to evaluate toxicity of newly developed antiviral drugs.

These stem-cell derived organoid models representing fetal organ development improve our ability to mimic the toxicity and efficacy of antiviral drugs through overcoming the two-dimensionality and genetic inaccuracies of cell lines. Of note, these cellular models are also associated with some limitations, such as difficulties with culturing, batch-effect or stem cell line effect on cell culture reproducibility [176] and unintended differentiation into mixed tissue type [177]. Most importantly, such organoid models lack vascular and immune system components [178], limiting their clinical translatability.

## 5. Conclusions

Although clinical trials are the gold standard for drug safety and efficacy testing, the predominant exclusion of pregnant women in such trials heightens the need for innovative research models to address these questions. The application of ex vivo models of the maternal-fetal interface has advanced our ability to study different stages of pregnancy without the need for continuously collecting fresh human tissues from ongoing pregnancies. Cellular models representing the human placenta and the developing fetus have advanced our ability to study drug toxicity and efficacy throughout pregnancy. The translational potential of the reproductive toxicity data generated from these models show great promise and warrant further investigation.

**Author Contributions:** Conceptualization, S.L.H., M.C.S., E.A.M., R.E.S., S.C., L.E.R., H.S., Y.J.Y. and R.G.; writing—original draft preparation, S.L.H., M.C.S.; writing—review and editing, E.A.M., R.E.S., S.C., L.E.R., H.S., Y.J.Y. and R.G. All authors have read and agreed to the published version of the manuscript.

**Funding:** This research was funded by New York Presbyterian-Weill Cornell Medical Center Alumni Council grant (R.G.), Weill Cornell Medicine- Department of Pediatrics Research support (R.G.) and Weill Cornell Medicine COVID 19 Research Grant (Y.J.Y., H.S., R.E.S.).

**Institutional Review Board Statement:** Not applicable.

**Informed Consent Statement:** Not applicable.

**Data Availability Statement:** Not applicable.

**Conflicts of Interest:** S.C. is the co-founder of Oncobeat, INC and is a consultant for Vesalius Therapeutics.

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
