# Peer review of "Human Maternal-Fetal Interface Cellular Models to Assess Antiviral Drug Toxicity during Pregnancy"

_2673-3897, doi:10.3390/reprodmed3040024_

Round 1

Reviewer 1 Report

The authors described several cellular models representing the placenta to assess antiviral drugs toxicity during pregnancy. Overall, the review was well written. However, the information on the approaches specific to antiviral drug toxicity is limited. It would be better to add more possible approaches specific to the antivirus drug study.

Author Response

We thank the reviewers for this suggestion. As requested, we have now discussed the previously used approaches for antiviral toxicity studies in placental and fetal cellular models in the following lines.

Lines 223-224: In one study, the effect of ZIKV antiviral Palmitoleate on trophoblastic apoptosis and endoplasmic reticulum stress was investigated [80].

Lines 360-363: In this study, the survival and proliferation of hESCs treated with RDV or GS-441524 was assessed and the ability of RDV, but not GS-441524, to decrease the proliferation of cells in a dose-dependent manner was indicated [139].

Lines 399-401: These organoid models were used to screen non-toxic antiviral drugs by evaluating their effects on cell survival, proliferation, and apoptosis [147-149] .

Reviewer 2 Report

The high risks for pregnancy’s evolution deriving from a particular viral disease, may result in serious health consequences for both the mother and the fetus. As in adult population antiviral drugs carry a long history, for now, there is a lack comprehensive safety and efficacy data for their use among pregnant women. For safety reasons, pregnant women are systematically excluded from therapeutic clinical trials fearing from potential fetal harm. The current approach limits testing new drug molecules on small animals which often do not accurately predict the human toxicological profiles of the tested drugs. The present material reviews the potential of human maternal-fetal interface cellular models in reproductive toxicity assessment of antiviral drugs. In the current situation, screening of drug candidates in physiologically relevant human maternal-fetal cellular models will be beneficial to prioritize selection of later safe antiviral therapeutics for clinical trials including pregnant women. It is a very interesting material and I congratulate the authors for gathering this data in such a comprehensive manner.

I suggest to add in the Lines 93-96 some information about the coagulation path that proved to be deteriorated in the context of Covid-19 Infection, this is not included between the mentioned mechanisms.

Author Response

We thank the reviewers for their interest in the topic of the review. According to the reviewers’ suggestion, we have added additional mechanistic information regarding complement activation and procoagulant effects in the placenta, leading to SARS-CoV-2 placentitis in line 96-98.

Lines 96-98: This placental damage is likely mediated by complement activation leading to increased cytokine activity and procoagulant effects [35].